# Anti-Aging Potential of Platelet Rich Plasma (PRP): Evidence from Osteoarthritis (OA) and Applications in Senescence and Inflammaging

**DOI:** 10.3390/bioengineering10080987

**Published:** 2023-08-21

**Authors:** James Vun, Neelam Iqbal, Elena Jones, Payal Ganguly

**Affiliations:** 1Leeds Institute of Rheumatic and Musculoskeletal Medicine, University of Leeds, Leeds LS9 7TF, UK; umjshv@leeds.ac.uk (J.V.); msjej@leeds.ac.uk (E.J.); 2Leeds Orthopaedic & Trauma Sciences, Leeds General Infirmary, University of Leeds, Leeds LS97TF, UK; 3School of Chemical and Process Engineering, University of Leeds, Leeds LS2 9JT, UK; n.iqbal2@leeds.ac.uk

**Keywords:** platelet rich plasma (PRP), aging, senescence, inflammation, inflammaging, anti-aging, rejuvenation, age-related changes, age-related diseases (ARDs), osteoarthritis

## Abstract

Aging and age-related changes impact the quality of life (QOL) in elderly with a decline in movement, cognitive abilities and increased vulnerability towards age-related diseases (ARDs). One of the key contributing factors is cellular senescence, which is triggered majorly by DNA damage response (DDR). Accumulated senescent cells (SCs) release senescence-associated secretory phenotype (SASP), which includes pro-inflammatory cytokines, matrix metalloproteinases (MMPs), lipids and chemokines that are detrimental to the surrounding tissues. Chronic low-grade inflammation in the elderly or inflammaging is also associated with cellular senescence and contributes to ARDs. The literature from the last decade has recorded the use of platelet rich plasma (PRP) to combat senescence and inflammation, alleviate pain as an analgesic, promote tissue regeneration and repair via angiogenesis—all of which are essential in anti-aging and tissue regeneration strategies. In the last few decades, platelet-rich plasma (PRP) has been used as an anti-aging treatment option for dermatological applications and with great interest in tissue regeneration for orthopaedic applications, especially in osteoarthritis (OA). In this exploration, we connect the intricate relationship between aging, ARDs, senescence and inflammation and delve into PRP’s properties and potential benefits. We conduct a comparative review of the current literature on PRP treatment strategies, paying particular attention to the instances strongly linked to ARDs. Finally, upon careful consideration of this interconnected information in the context of aging, we suggest a prospective role for PRP in developing anti-aging therapeutic strategies.

## 1. Introduction

Advancing age and age-related changes have presented challenges in healthcare across the globe ranging from increased vulnerability towards diseases to reduced quality of life (QOL). According to the World Health Organisation (WHO), by the year 2030, every 1 in 6 person will be over 60, and by the year 2050, there will be over 2 billion people aged 60 and above [1]. Biologically, aging has been defined as accumulating several damages at the cellular level over time, potentially resulting in reduced physical and mental abilities. Therefore, it often results in various age-related diseases (ARDs), accelerating the natural progression towards death [2]. Since advancing age is closely associated with ARDs and global health challenges, the WHO predicts that the increasing aging population will present economic and social challenges worldwide [1].

The last three decades have witnessed an increasing number of studies investigating age-related changes at the cellular level. Re-defining the underlying causes of these changes was the 2013 article by Lopez-Otin et al. describing the nine hallmarks of aging [2]. They outlined nine factors and categorised them as primary hallmarks—that cause the damage (genomic instability, telomere attrition, epigenetic alterations, loss of proteostasis); antagonistic hallmarks—that are the responses to the caused damage (deregulated nutrient sensing, mitochondrial dysfunction and senescence); and integrative hallmarks—that eventually result in the phenotypes of aging (stem cell exhaustion, altered intercellular communication) [2]. However, since then, investigations of age-related changes have amplified due to the increased interest among researchers. More recently, the nine hallmarks of aging were updated as the ‘new hallmarks of aging’ at the 2022 Copenhagen Aging meeting [3]. The added factors included age-related inflammation—also referred to as inflammaging, compromised autophagy, altered mechanical properties, microbiome disturbance and dysregulation of RNA processing.

Several conditions and ARDs have been found to severely impact the elderly, including neurodegenerative diseases leading to cognitive impairment, hearing loss, cardiovascular complications and reduced mobility due to musculoskeletal (MSK) alterations [4]. Among these, hip and knee osteoarthritis (OA) are known to be the most significant contributors to global disability, with the likelihood of its burden being underestimated [5]. Pain, rigidity in the joints, inflammation and difficulties in movement are well-recognised symptoms of OA. Despite the continuous efforts in OA research, the usual approaches include pain management and/or invasive surgeries. At the cellular level, various factors contribute towards OA; however, advancing age, age-related inflammation and senescence have been identified as critical factors influencing the disease progression [6,7,8,9]. Senescence is the irreversible arrest of cellular growth and proliferation and contributes towards aging and age-related changes. Senescent cells (SCs) produce senescence-associated secretory phenotype (SASP) that are also closely linked to contributors of several ARDs. SASP is also known to fuel the chronic, systemic-low grade inflammation in aging, that is, ‘inflammaging’, a significant risk factor contributing towards ARDs [10].

Interestingly, there has been an increase in the use of platelet rich plasma (PRP) for treating ARDs like OA [11,12,13] in the last three decades. PRP has been described as, ‘a fraction of blood plasma containing platelets in a higher concentration than at baseline blood levels’ by Sanchez et al. [14]. Some studies have described it as a fluid with a platelet concentration of at least 1 × 10^6^ platelets/mL in 5 mL of plasma, known to contain 3 to 5-times the growth factor concentration and associated with enhancement of healing [15,16]. Over 1100 proteins have been found in PRP, including growth factors, enzymes and messenger molecules [17]. These contribute towards processes like proliferation, differentiation, angiogenesis and matrix modelling [18,19]. The treatment approach involves injection of autologous PRP containing these potent growth factors into the patient, and is not known to have any adverse effects. The primary application for PRP-based therapy relies on the rationale that concentrated platelets injected at the injury site may initiate tissue repair via the multiple growth factors residing inside PRP [20]. The procedure has now been outlined as a guideline by the National Institute for Health and Care Excellence (NICE), UK [21], to alleviate pain and relieve symptoms in OA patients.

While several trials and investigations have investigated the effect of PRP on OA treatment [22,23], there has also been an increased interest in using PRP in other applications. These include treatment of androgenic alopecia and hair regrowth [24,25], bone and spine regeneration [26,27,28], dentistry [29] and skin regeneration [17,30]. Along with several other growth factors and proteins, PRP is also known to contain tissue inhibitors of metalloproteinase 2 (TIMP2) and growth differentiation factor 11 (GDF11) proteins that play a critical role in tissue regeneration and repair. Thus, these are also referred to as the anti-aging proteins in PRP, which have been reported to decline with advancing age and in diabetes mellitus [31].

Last year, we worked with mesenchymal stromal cells (MSCs) from OA patients before and after PRP injections. MSCs from older OA patients had a higher increase in their proliferative capacity. They also demonstrated higher protection from oxidative stress as indicated by senescence assay [32]. Reported investigations of the anti-inflammatory potential of PRP [33] and the use of PRP to combat senescence [34,35] further indicate the potential of PRP in use beyond OA in other ARDs for tissue regeneration. Considering our recently published study [32], the increase or use of PRP in OA and the current literature available outlining the impact of PRP in therapeutic applications mentioned above, we hypothesised that PRP potentially has anti-aging properties that can be further exploited for age-related changes and the treatment of several ARDs. This review delineates the current evidence to link aging, senescence, inflammaging and ARDs, with OA being our reference ARD. We then discuss the potential of PRP being applied for their anti-aging properties in aging in general as well as in several other ARDs.

## 2. Senescence and Inflammaging

### 2.1. Senescence

The term senescence was first used in 1961 and described as an irreversible loss of proliferative capacity in somatic cells [36]. Since then, we have gathered a much better understanding of the biological significance of senescence. While the mechanism of senescence is crucial for biological events like tumour suppression [37], it has increasingly been associated with tumour progression, damage accumulation, aging and several ARDs [38]. SCs are identified by the loss of their ability to proliferate, by their larger-than-normal size, by the positive stain for senescence-associated beta-galactosidase (SA-β-gal) and by higher expression of transcription factors like p16 and p53 [38,39]. It may be caused due to various factors or stressors contributing towards SCs; however, DNA damage response (DDR) remains among the most well-explored stressors for senescence [40,41,42].

Cellular senescence depletes tissues of the progenitor of stem cells, leading to one of the hallmarks of aging: stem cell exhaustion, that is, a decline in the number of stem cells [43,44]. This exhaustion eventually leads to compromised tissue repair and regeneration [45] and damage in the surrounding tissues [46,47]. Concerning OA, the impact of senescence is well acknowledged and is often used for dissecting the pathways involved in the disease progression [6,7,8,9,48,49,50]. Additionally, clearance of SCs in mice using an inhibitor of anti-apoptotic molecules (ABT263) or by using transgenic mice resulted in the rejuvenation of hematopoietic stem cells (HSCs) [51] and relieved pain and increased cartilage development [52], respectively. A similar observation was made by treating aged mice with a senolytic drug (Navitoclax), where the drug eliminated senescent cardiomyocytes, improved myocardial modelling and overall survival after myocardial infarction [53]. These observations have made senescence an interesting target for potential anti-aging therapies.

SCs produce factors known as SASP, which is how the SCs contribute towards aging and ARDs. SASP includes a mix of pro-inflammatory cytokines, chemokines, lipids, matrix-metalloproteinases (MMPs) and matrix-degrading molecules, accelerating cellular damage and advancing aging and ARDs [54,55]. The presence of the pro-inflammatory cytokine and the chemokines inhibit functions like proliferation and differentiation and cause cellular damage [38]. SASP has been found to increase with age with a dynamic and long-lasting program for SCs. Gorgulis and colleagues collated and described several factors that may be used as a guideline for senescence markers [56]. It plays a significant role in SCs that ultimately contributes towards accelerated aging and ARDs via the process of cellular senescence [57] and, as outlined earlier, also towards inflammaging [10].

### 2.2. Inflammaging

Inflammaging describes the upregulated inflammatory state of tissues that progresses with advancing age and has been added to the scientific dictionary only in the last two decades [58]. It is identified by an increase in the levels of several pro-inflammatory cytokines of interleukins 1, 6 (IL-1, IL-6) and tumour necrosis factor (TNF), which are all well recognised in inflammatory diseases [59,60,61]. Inflammaging is believed to result from continuous lifetime exposure to clinical and non-clinical infections and non-infectious agents causing irreversible cellular and molecular damage, which is not always clinically evident. However, its exact mechanisms are yet to be underpinned, although inflammaging is believed to share similarities in pathways with inflammation in diseased conditions [61,62].

Inflammaging has been projected to withhold significant risk towards morbidity and mortality in the health of the elderly, as it shares the inflammatory pathogenesis with a number of ARDs and other degenerative joint diseases [62,63,64]. For example, in OA, both inflammation and cellular senescence are key contributors to the disease progression. Senescent osteocytes increase in OA and also increases the receptor activator of nuclear kappa-B ligand (*RANKL*)-related rate of bone resorption [65,66]. The process subsequently triggers inflammation and the related stages, encompassing the continuous effort for healing. It is hypothesised that this process intensifies the severity of the condition [67,68]. Considering OA as our reference for an ARD for this article, it appears closely integrated with inflammaging, as are other ARDs [69], making inflammaging an increasingly unavoidable factor in aging and regenerative research. Thus, it is no surprise that inflammaging has, like senescence, also been hypothesised to be a potential target towards anti-aging strategies [63,70].

## 3. Platelet-Rich Plasma (PRP)

PRP has been used in numerous studies for its regenerative potential, especially in OA and has been discussed in great detail in other publications [12,23,32,71,72,73,74,75]. This section outlines our current knowledge of PRP and its properties that make it an attractive agent for use in these studies. We acknowledge the ambiguity in its terminology used in the literature and then focus on the current application of PRP by outlining its clinical relevance. We then summarise the results of these applications to critically assess if PRP, in reality, contributed to regenerating the tissues and disease management (Table 1, Section 4.1).

### 3.1. Properties and Contents of PRP and Platelet-Derived Biologics

When harvested with anticoagulants, peripheral blood can be centrifuged to produce two distinct fractions: platelet rich plasma (PRP) and platelet poor plasma (PPP). PRP, obtained from a patient’s peripheral blood, is a rich source of autologous growth factors and anti-aging proteins [31,76]. Noteworthy, not all PRP described in the literature are the same regarding physical properties and growth factor contents [20]. ‘Non-activated’ PRP is produced following peripheral blood centrifugation and delivered in liquid form. Therefore, the release of growth factors from non-activated PRP occurs in vivo through intrinsic biological activation by the host ‘on demand’. On the other hand, the release of growth factors from ‘activated’ PRP was induced ex vivo prior to administration. These were commonly induced through physical disruption, lysis or chemical activation. Physical disruption, usually through repeated freeze–thaw cycles or ultrasound sonification, results in platelet lysis and the production of acellular ‘platelet lysate (PL)’ abundant in growth factors [77]. Chemical activation with calcium salts or thrombin leads to (i) platelet degranulation and release of a plasmatic fraction full of growth factors, known more commonly as platelet releasate, a plasma rich in growth factors, or platelet release supernatants; and (ii) fibrinogenesis and the formation of a platelet-rich clot—platelet gel [78].

PRP remains loosely used in the literature as an all-encompassing term for any platelet-derived biologic. A recent systematic review highlighted the importance of accurately describing platelet-derived biologic, physical properties, formulation and contents to enable accurate comparison between studies [79]. The shortcomings of the existing PRP classification systems are highlighted by the lack of standardised protocols and definition for centrifugation and preparation, cellular components such as platelet concentration, red and white blood cell counts, and the procedure for platelet activation [80,81,82,83,84,85]. However, in 2020, Kon et al. suggested a novel coded classification system for PRP. The code includes a sequence of six digits grouped in pairs indicating parameters of platelet composition, purity, and activation: N1-N2-N3-N4-N5-N6. While it is yet to be accepted as a standardised method, it would provide essential details in a simplified manner [13]. PRP and related products have been given several names, such as leukocyte rich-PRP (LR-PRP), leukocyte poor-PRP (LP-PRP), platelet-rich fibrin (PRF) and pure-PRP (pPRP) based on the formulation prepared for the particular study. Though these terminologies are beyond the remit of this article, Everts et al. have provided an exhaustive review of the subject [20].

### 3.2. Current Applications of PRP

Due to its ease of harvesting, as well as being autologous and readily available, PRP has gained popularity and is clinically applied in treating acute fractures, fracture non-union, diseased or inflamed tendons (tendinopathies), OA and wound healing. A recent systematic review and meta-analysis on 10 randomised controlled trials (RCT) with 652 patients identified PRP as being able to reduce fracture healing time, enhance bone mineral density and decrease the risk of revision surgeries [86]. Nonetheless, due to the absence of randomised controlled trials (RCTs) evaluating the effectiveness of PRP in treating fracture non-union, its incorporation into the standard surgical management of fracture non-unions has not yet been realised [87].

PRP has also been used for the treatment of spinal diseases. Intradiscal injection therapy of PRP for degenerative disc disease is safe and improves pain, disability and QOL [88]. Numerous in vitro investigations of PRP in spinal diseases have been reviewed extensively by Kawabata et al., indicating that PRP can potentially be a breakthrough treatment in spine-related diseases. However, they also stated that clinical outcomes of PRP applied in spinal fusion surgery and spinal cord injury have been mixed, with studies differing in their final results [27] . Ultimately, they summarise the need for larger RCTs with careful patient selection and PRP component considerations to establish PRP as an evidence-based treatment [27]. With regards to tendon pathologies, Masiollo et al. in their recent systematic review and meta-analysis, found no significant differences in terms of pain scores and functional outcome scores when comparing the use of PRP injections against other treatments (e.g., steroid injection, local anaesthetic, MSC injection or conservative non-injective interventions) when applied in lateral epicondylitis, plantar fasciitis, achilles tendinopathy or rotator cuff tendinopathy [89].

As mentioned earlier, the application of PRP in OA has been widely studied. Studies have explored the application of PRP alone, the combinatory use of PRP with other biologics, or PRP as an adjunct to surgical procedures in the treatment of OA. A recent meta-analysis assessing six RCTs compared the use of PRP (*n* = 338) against hyaluronic acid (HA) (*n* = 323) in knee OA, concluded that PRP is superior over HA in improved Western Ontario McMaster Universities OA Index (WOMAC) and Physical Functional Scores at 12 months, while there was no difference in the International Knee Documentation Committee (IKDC) and EuroQol—visual analogue scales (EQ-VAS) Scores [90]. In their meta-analysis of six RCTs (*n* = 493 cases), Zhao et al. identified the combined application of PRP with MSC (PRP+MSC) to significantly reduce VAS scores at 6 and 12 months when compared against control, HA injection or PRP alone; and WOMAC scores to be more effectively reduced at 3 and 6 months following MSC+PRP treatment. Knee injury and OA outcome score (KOOS) at 12 months indicated no difference between MSC+PRP and control groups [72]. Another systematic review similarly reported on the longevity of pain relief and improvement of functional outcomes following PRP injection when compared against PPP, corticosteroids and HA [75].

The use of ‘orthobiologics’ in the treatment of OA, with PRP being one of the more popular blood-based derivatives, has been the recent focus of the European Society of Sports Traumatology, Knee Surgery and Arthroscopy (ESSKA) Orthobiologics Initiative (ORBIT) steering group [91]. The ESSKA ORBIT group drew attention to and challenged the focus of the current literature, which is often limited to reporting the pain-relieving effects and functional improvements following injectables in OA (i.e., orthobiologics such as PRP, corticosteroids, viscosupplementation such as HA). The ORBIT steering group further highlighted the importance of looking at whether PRP contains any disease-modifying effects in the pathophysiology of OA, leading to the group’s systematic review of 44 pre-clinical studies with 1251 animals. The systematic review demonstrated PRP, when used to treat OA in animal models, has both clinical (80% of studies) and disease-modifying effects (68% of studies) [92].

The observed disease-modifying effects of PRP included a reduction in cartilage disease progression and synovial inflammation in OA, along with changes in biomarker levels following PRP administration. PRP was noted to significantly reduce synovial fluid concentrations of biomarkers related to inflammation including cartilage oligomeric matrix protein or (COMP), serum and synovial fluid levels of TNF-a, IL-1β, and prostaglandin E2 (PGE2) levels, and mRNA expression of MMPs [92]. The authors found that PRP significantly increased serum levels of platelet-derived growth factor-A (PDGF-A) and vascular endothelial growth factor (VEGF) and reduced the OA research society international (OARSI) score. Interestingly, VEGF has been shown to be detrimental to cartilage repair in OA [93,94] with further evidence in studies that focussed on factors like using anti-VEGF antibody as a therapeutic approach towards post-traumatic OA [95] or by VEGF attenuation in PRP which enhanced repair in rat OA cartilage [96]. Another study demonstrated that the use of Alendronate (used for the treatment of osteoporosis) in animals, via the inhibition of the nuclear factor kappa-B (NF-κB) signalling pathway [97]. These studies indicated a decrease in the production of inflammation, inflammatory markers, attenuation of cartilage damage and delay in OA progression with the use of PRP [91,97]. All of these parameters are closely related to senescence, a hallmark of aging. However, none of these studies explored the direct association with senescence or any anti-aging potential of PRP.

In contrast, the anti-aging potential of PRP has found extensive applications in fields like dermatology, plastic surgery and overall wound healing [98]. It has been utilised in treating androgenic alopecia, with multiple randomised controlled trials (RCTs) reporting increased hair count, density and thickness among patients who underwent PRP treatment [99,100,101,102]. However, no correlation has been established between platelet count, PDGF, epidermal growth factor (EGF), VEGF levels and clinical improvement (hair count, hair density) [102]. While some meta-analyses have noted improved healing rates, faster healing times and better scar evaluation scores in burn wounds treated with PRP [103], others have found that PRP offers no additional advantage over control in reducing wound infection rates in cases of sternal wounds [104] and burn wounds [103]. Additionally, an RCT that examined the application of autologous leukocyte PRP on the wounds of patients who underwent total hip replacement suggested that leukocyte PRP might expedite complete wound healing, implying a potential role for EGF receptor agonists in promoting wound healing [105].

## 4. Anti-Aging Potential of PRP for ARDs

### 4.1. Current Evidence in Literature

The evidence outlined above suggests that while there have been cases when PRP may not have indicated additional benefits in tissue regeneration in comparison to control/placebo; there are, however, multiple indications of the potential of PRP for hair regeneration [102], skin aging [106], wound healing [107], joint repair with and without the combination of MSCs [28,72,108]. We know that inflammation, inflammaging, tissue damage and senescence contribute to aging and ARDs. With the advancement of age, it is well-established that an accumulation of damage leads to cellular senescence and secretion of SASP [2,37,38]. SASP is detected by the presence of pro-inflammatory cytokines and low-grade inflammation (or inflammaging), which eventually leads to the declined ability of tissues to regenerate and repair [10,61,62,69], especially in the case of OA [6,8,48,49,64]. Thus, if PRP has increasingly been used for tissue repair and regeneration with an increasing body of evidence for senescence and inflammation (Table 1), it merits investigation of the anti-aging potential of PRP as well (Figure 1).

A study by Cole et al. in 2017 found that while PRP treatment was comparable to HA treatment regarding WOMAC pain score, their observations and results found that PRP potentially played an anti-inflammatory role for *n* = 49 OA patients [12]. The anti-inflammatory effect of PRP was also observed in a study by Zhang et al. for in vitro rabbit cells and in vivo mouse models, and it was estimated to do so via hepatocyte growth factor (HGF) [109]. Apart from these, another animal study with old mice indicated that intravenous (IV) administration of PRP increased locomotion, enhanced cognitive function and reduced anxiety and depression-like symptoms [35]. More interestingly, Cakiroglu et al. investigated the effect of autologous intra-ovarian PRP injections in women suffering from primary ovarian insufficiency (POI) to see if PRP improved in vitro fertilisation (IVF) outcomes. Post-PRP treatment, 7.4% of women conceived simultaneously and 64.8% developed antral follicles and were able to attempt IVF [110].

Table 1 below outlines these and similar studies that indicate results for the regenerative and angiogenic properties, tissue regenerative abilities, anti-inflammatory role, analgesic and anti-senescence effects of PRP, indirectly connecting it all with the anti-aging potential of PRP.

**Table 1 bioengineering-10-00987-t001:** List of studies investigating the anti-aging potential of PRP via several parameters.

PRP Used	Model	PRP Details	Target Disease/Organ	Senescence/Aging-Related Parameters Tested	Main Age-Related Findings	Ref.
Autologous IA PRP	In vivo: prospective, double-blinded, comparative RCT between PRP and HA	Leukocyte poor, single spin, PRP average volume of 4 mL	Knee OA	Inflammatory/catabolic markers by ELISA and pain scores (WOMAC, IKDC, VAS)	PRP indicated significantly better outcomes at weeks 24 and 52. Additionally, PRP treatment decreased the secretion of two pro-inflammatory cytokines, i.e., IL-1β and TNF-α	[12]
Human PRP releasate coated onto porous collagen-glycosaminoglycans (GAG) scaffolds	In vitro: human bone marrow mesenchymal stromal cells (MSCs) and human pooled endothelial cells	Platelet concentration of 1 × 10^6^/mL formulated into PRP releasate	Skin wound healing application	Proliferation, migration, angiogenesis, vascularisation, macrophage polarisation	PRP-coated scaffold exhibited enhanced angiogenic and vascularisation potential. It also indicated a lower release of pro-inflammatory markers when exposed to an inflammatory environment	[30]
Autologous IA and IO PRP for patient treatment before cell isolation	Ex vivo: human bone marrow MSCs	Liquid PRP IA: 8 mL and IO: 10 mL volumes, respectively	Hip OA	MSC proliferation, SA-β-gal, gene expression	Enhanced MSC proliferation and stress resistance capacities in older MSC donors treated with PRP	[32]
Healthy male New Zealand white rabbit PRP; cell culture treated	In vitro: murine dermal fibroblasts	1%, 5% and 10% PRP diluted in 1%FBS for in vitro cultures	Senescence associated phenotypes in photo aging	Cell morphology, SA-β-gal, cell cycle arrest, ECM, ROS	PRP prevented cell cycle arrest, reduced SA-β-gal, increased ECM collagen and decreased MMPs. PRP may counteract cell senescence in MSK	[34]
Allogenic, IV PRP	In vivo: senescent female BALB/c mice (16–18 months old); *n* = 46	Not provided	Cognitive aging in senescent mice	Behavioural and cognitive: open-field, elevated plus maze, tail suspension, and Morris water maze test	PRP increased locomotion, improved memory, reduced anxiety and depression-like behaviour in old mice; enhanced cognitive function with PRP	[35]
Enhanced PRP (ePRP) powder prepared by freeze-drying PRP from human patients	In vitro: human dermal fibroblasts	Platelet count: 1 × 10^9^/mL lyophilised powder reconstituted with 1 mL media	Wound healing in fibroblasts in vitro	Proliferation, wound healing, qPCR, intracellular ATP measurement, SA-β-gal, metabolic flux	ePRP stimulates wound healing by releasing growth factors, increased anti-oxidant production, halts the senescence progression of fibroblasts by activating *SIRT1* expression	[107]
PRP prepared from rabbit/mouse used for the study	In vitro and in vivo: rabbit tendon cells in vitro studies and mice for in vivo studies	In vitro: 10%PRP and in vivo: 10 mL PRP	Injured tendons	In vitro—COX-1, COX-2, mPGES levels by gene expression, Western blotting and immunostaining; in vivo—immunohistochemistry	PRP treatment suppressed tendon cell inflammation in vitro and tendon inflammation in vivo, at least partially mediated by HGF in PRP	[109]
Intra-ovarian delivery of autologous PRP	In vivo: women with primary ovarian insufficiency (POI)	2–4 mL of PRP was injected underneath the ovarian cortex	Ovarian stimulation and IVF outcome in women with POI	Antral follicular count, follicle-stimulating hormone, IVF, clinical pregnancy	PRP treatment aided conception in women suffering from POI, with 7.4% of the women being able to conceive spontaneously after PRP treatment	[110]
Human PRP is used to treat NIH3T3 cells to be transplanted in vivo	In vivo: genetically modified NIH3T3 embryonic fibroblasts with an enhanced green fluorescent protein (NIH3T3-G) differentiated into osteoblasts transplanted into variectomized senescence-accelerated mouse prone substrain 8 (OVX-SAMP8 mice)	Not provided	Bone regeneration applications in osteoporosis and conditions associated with accelerated aging	Molecular imaging and immunohistochemistry	(PRP/NIH3T3-G) engraftment prevented the development of osteoporosis, and the life span of OVX-SAMP8 mice receiving PRP/NIH3T3-G transplantation was significantly prolonged	[111]

PRP—Platelet rich plasma, IA—intra articular, IO—intra osseous, IV—intra venous, RCT—randomised controlled trial, HA—hyaluronic acid, BALB/c—Bagg Albino mice, ELISA—enzyme-linked immunosorbent assay, WOMAC—Western Ontario and McMaster Universities Osteoarthritis index, IKDC—International knee documentation committee, VAS—Visual analogue scale, SA-β-gal—senescence-associated beta-galactosidase, ECM—extracellular matrix, ROS—reactive oxygen species, qPCR—quantitative polymerised chain reaction, ATP—adenosine triphosphate, COX-1,2—Cyclooxygenase 1, 2, mPGES—microsomal prostaglandin e-synthase, IVF—in vitro fertilisation, IL-1β—interleukin 1-beta, TNF-α tumour necrosis factor-alpha, MSK—musculoskeletal, *SIRT1*- Sirtuin 1, HGF—hepatocyte growth factor.

### 4.2. Potential Mechanism of Action (MOA)

PRP has already been established for its anti-aging properties in the field of dermatology and skin care, as outlined earlier. However, the exact mechanism underpinning the effect of PRP remains to be established. With respect to skin aging, it has been reported that injecting PRP into aging skin activates a series of cellular reactions, and cellulose, fibronectin and vitronectin from the PRP aggregate with the growth factors and act locally at the injected site [18]. These proteins potentially act as scaffolds for nascent cells and tissues to advance the repair of aging skin by inducing DNA synthesis and promoting corresponding gene expression at the cellular level [17,112,113] Use of PRP on aging skin resulted in increased reticular dermis thickness by deposition of collagen and elastin fibres [113], and an in vitro model indicated that PRP treated aging skin ameliorated photo aging of the skin via inhibition of MMP-1 and tyrosinase upregulation and by inducing tropoelastin and fibrillin expressions [17].

Regarding regenerative medicine applications of PRP other than dermatology, our knowledge of its action is increasing, but the exact mechanisms are yet to be elucidated [114]. It has, however, been established that platelets remain inactive inside PRP until injected into the injury site, where upon contact with tissue factor—platelets change shape and develop pseudopods to promote platelet aggregation [20]. The activated platelets at the site of injury function via PDGF, VEGF, TGF-b and EGF to promote cellular processes leading to angiogenesis, regeneration, proliferation and bone healing [115].

Platelets are anucleate blood cells produced mainly by megakaryocytes in the bone marrow that are 2–4 mm in diameter with a life span of 7–10 days, with approximately 10^11^ platelets produced daily [116]. In contrast, fibroblasts are reported to have a life span of 270 days in vitro [117], osteoblasts have a life span of 1–200 days, bone lining cells may live up to 1–10 years, and osteocytes can live up to 50 years [118]. Considering the long life span of these bone cells, it can be acknowledged that they accumulate considerable amount of cellular damage leading to SCs and SASP. On the other hand, platelets renew every 7–10 days, indicating that senescence may not be as deep seated in them. The release of growth factors and cytokines in platelet granules accelerates tissue repair, and this process is further enhanced once combined with fibrinolytic systems [119]. This may be one of the mechanisms by which PRP portrays regenerative and, ultimately, anti-aging properties.

## 5. Limitations, Conclusions and Future Directions

A critical shortcoming of PRP is its inconsistency, which applies not just to the nomenclature (such as LR-PRP, LP-PRP and so forth) but also to the required formulation, i.e., concentrate, gel, etc., to utilise its benefits to its highest potential. This problem is typically mitigated by adjusting the formulation to suit the specific requirements of each project. Nevertheless, standardising these formulations could definitively establish the influence of PRP, particularly in exploring its impact on aging and ARDs. Another difficulty lies in navigating the intricate network of age-related alterations. Aging itself is exceedingly complex, and now, with 12 recognised hallmarks, the number of aspects requiring exploration to comprehend the anti-aging properties of PRP continues to expand. Before initiating the investigation, each study or trial must judiciously select which hallmarks or parameters align most appropriately with their research goals. Furthermore, considering the existing studies that have observed improvements, and others that have not with PRP treatment [89], it is plausible that for effective PRP therapies, autologous PRP will need to be specifically tailored for each patient or case, resonating with the concept of precision medicine [120].

Ideally, OA clinical trials or studies investigating the effect of PRP must include tests for quantifying senescence and its markers as regular screening. Trials focussing on IO and/or IA PRP injections would have a higher ability to locally impact more SCs as these methods would reach closer to the longest living cells in the bone that would have accumulated considerable SCs and SASP. Serum levels of pro-inflammatory cytokines by ELISA from these patients, along with quantification of SC from inside the bone marrow cells using SA-b-gal assays, are feasible methods [32] and would help shed light on the anti-senescence effect of PRP treatment in OA, which is an ARD. Additionally, with the previous literature indicating that factors/proteins/microenvironment from younger donors can potentially regenerate tissues in older donors [43,121,122,123], it will be worth exploring the application of allogenic PRP from younger donors to patients with OA and eventually, in other ARDs.

Although we are not alone in introducing the concept of PRP as a potential anti-aging solution for age-related diseases (ARDs) [20], to the best of our knowledge, this is the inaugural perspective that provides evidence linking the use of PRP to both aging and age-associated factors of inflammation, inflammaging and senescence that ultimately lead to ARDs. PRP’s functionalities span numerous applications, including analgesics, anti-inflammatory, anti-senescent and angiogenesis (Table 1). Based on the currently available evidence, this perspective is unique in its exclusive focus on PRP’s anti-aging potential. While PRP has not always been shown to enhance tissue regeneration, we believe that with the appropriate optimisation of concentration and modification of its formulations, PRP can intervene in age-related transformations, thereby slowing the progression of ARDs.

In the future, experimental medicine that optimises PRP concentration for patient use could elucidate the mechanisms behind PRP’s anti-aging effects. Double-blinded RCTs assessing PRP’s impact on ARDs will assist in identifying the phenotypes that respond positively to PRP exposure. Long-term investigations and advanced examinations at the cellular and genetic levels, using state-of-the-art techniques such as next-generation sequencing, will help reveal new pathways that could serve as potential therapeutic targets in the future [124].

## Figures and Tables

**Figure 1 bioengineering-10-00987-f001:**
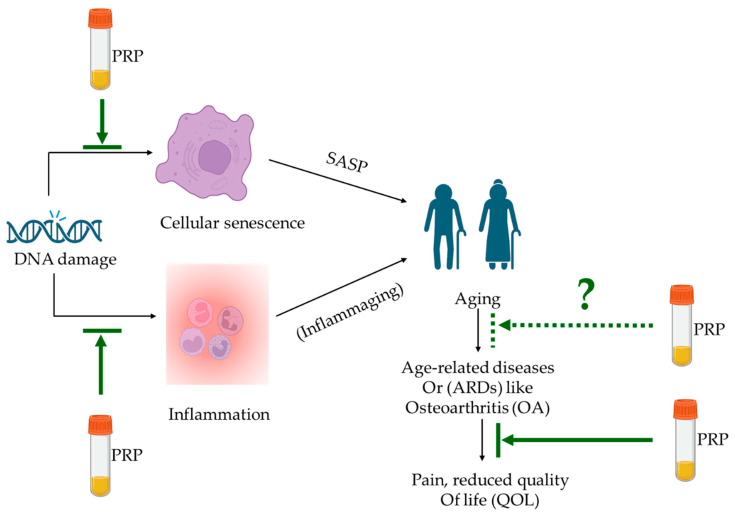
**Anti-aging potential of PRP.** DNA damage leads to cellular senescence and inflammation, which, with advancing age, accumulates SASP and inflammaging factors, respectively. Both contribute towards aging and age-related changes, specifically towards age-related diseases (ARDs) that impact the quality of life (QOL). PRP intervention has been shown to alleviate pain and disease progression in OA and reduce inflammatory markers and senescent cells (indicated in solid green lines), all of which are either precursors of aging or the effect of age-related changes. Investigations focused on the anti-aging property of PRP for age-related diseases (excluding those aimed at cosmetic applications) are yet to be explored and are indicated in the green dotted line.

## Data Availability

Not applicable.

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
