# Peer review of "Anti-Aging Potential of Platelet Rich Plasma (PRP): Evidence from Osteoarthritis (OA) and Applications in Senescence and Inflammaging"

_bioengineering, 2023, doi:10.3390/bioengineering10080987_

Round 1

Reviewer 1 Report

The authors have summarized the recent advance of PRP in OA, and the role of PRP in OA disease modifying and the potential anti-ageing. 

This review is comprehensive and of interest. I believe it is appropriate to be published. 

none

Author Response

Please see the file attached

Reviewer 2 Report

1. The reference format does not seem to fit this journal.

2. The line spacing seems to be inconsistent. Check the format of the overall manuscript.

3. Line 20: PRP: Full name is required.

4. Line 66: Osteoarthritis-->osteo~

5. Line 81-84: PRP, by definition, 1X10^6~ : This is not the definition of PRP. rewrite the sentence.

6. It would be better to additionally indicate the amount of PRP used (volume, frequency, number, concentration, etc.) in Table 1. Additionally, indicate the target disease or organ in Table 1.

Reviewer 3 Report

References: They are not properly formatted.  Many references do not have last names of the authors and page numbers.   

Page 3: “PRP, by definition, is fluid with a platelet concentration of at least 1x106 platelets/ml in 5ml of plasma, known to contain 3 to 5-times the growth factor concentration and is associated with enhancement of healing [14, 15].” – The volume is not included in the definition of the PRP. The unit of platelet concentration should be corrected to platelets/microliter (not milliliter). The platelet concentration in the initial definition was 1,000,000 platelets/microliter, but many recent studies have used PRP with a platelet concentration below that. I would recommend using a broader PRP definition as a supraphysiological concentration of platelets in plasma.

Page 3: “These include treatment of androgenic alopecia and hair regrowth [23, 24], regenerative medicine and repair [25], dentistry [26] and skin regeneration [16, 27].” – The authors need to provide more specific applications/areas for “regenerative medicine and repair”.

Page 3: “Older OA patients responded better for MSC proliferative and stress-resistant capacities [29].” – This sentence is unclear. 

Page 3: “Considering our recently published study, the increase or use of PRP in OA and the current literature available outlining the impact of PRP in therapeutic applications mentioned above, we hypothesised that PRP potentially has anti-aging properties that can be further exploited for age-related changes and the treatment of several ARDs.” – A reference is needed for the authors’ recently published study. It seems there are some typos that need authors’ attention. 

Page 8: “PRP was noted to significantly reduce synovial fluid known as cartilage oligomeric matrix protein or (COMP), serum and synovial fluid TNF-É£, IL-1β, and prostaglandin E2 (PGE2) levels, and mRNA expression of metalloproteinases [89].” – It needs to be re-written.

Page 8: “PRP significantly increased serum levels of platelet-derived growth factor-A (PDGF-A) and vascular endothelial growth factor (VEGF), which play a protective role.” – More explanation is required about “a protective role” of PDGF-A and VEGF. Previous studies have shown that angiogenesis is detrimental to the cartilage repair and VEGF plays a critical role in OA progression. Ref) Enomoto H. et al., Am J Pathol 2003;162(1):171, Nagai T. et al., Arthritis Res Ther 2014;16(5):427, Lee J.S. et al., Biomater Sci 2022;10(9):2172

Page 8: “Additionally, an RCT that examined the application of autologous leukocyte PRP on the wounds of patients who underwent total hip replacement suggested that L-PRP might expedite complete wound healing, implying a potential role for EGF receptor agonists in promoting wound healing [99].” – Definition of L-PRP?

Figure 1: The PRPs shown between “DNA damage” and “Inflammation”/“Cellular senescence” may cause confusion. It looks like PRP lead “DNA damage” to “cellular senescence” or “inflammation”. 

Section 4.2: “On the other hand, platelets renew every 7-10 days, indicating that senescence may not be as deep seated in them and thus portray regenerative and, ultimately, anti-aging properties.” – Any further explanation? The effect of PRP comes from the cargo inside activated or disrupted platelets, not from non-senescent platelets. 

There are a few minor grammar errors and typos.  Please read thoroughly and make changes as needed. 

Round 2

Reviewer 1 Report

the authors have made appropriate change.

Reviewer 2 Report

The manuscript improved. My comments and minor errors have been addressed.